# A Multi-Medium Analysis of Human Health Risk of Toxic Elements in Rice-Crayfish System: A Case Study from Middle Reach of Yangtze River, China

**DOI:** 10.3390/foods11081160

**Published:** 2022-04-16

**Authors:** Hui Zhou, Tao Ge, Hui Li, Ting Fang, Huaiyan Li, Yanhong Shi, Rong Zhang, Xinju Dong

**Affiliations:** 1Anhui Province Key Laboratory of Farmland Ecological Conservation and Pollution Prevention, College of Resources and Environment, Anhui Agricultural University, Hefei 230026, China; 2018046@ahau.edu.cn (H.Z.); 18156069713@126.com (H.L.); ahnydx@foxmail.com (Y.S.); leehui8@aaas.org.cn (R.Z.); 2Anhui Research Institute of Geological Experiment, Hefei 230001, China; get810@126.com; 3Key Laboratory of Freshwater Aquaculture and Enhancement of Anhui Province, Fisheries Research Institute, Anhui Academy of Agricultural Sciences, Hefei 230031, China; fangting_candy@163.com; 4Department of Chemistry, University of Louisville, Louisville, KY 40292, USA; x0dong03@louisville.edu

**Keywords:** rice-crayfish system, food, toxic elements, health risk assessment, Monte Carlo simulation

## Abstract

Rice-crayfish system has been extensively promoted in China in recent years. However, the presence of toxic elements in soil may threaten the quality of agricultural products. In this study, eight toxic elements were determined in multi-medium including soil, rice, and crayfish from the rice-crayfish system (RCS) and conventional rice culture (CRC) area. Crayfish obtained a low level of toxic element content, and mercury (Hg) in rice from RCS showed the highest bioavailability and mobility. Health risk assessment, coupled with Monte Carlo simulation, revealed that the dietary exposure to arsenic (As) and Hg from rice and crayfish consumption was the primary factor for non-carcinogenic risk, while Cd and As were the dominant contributors to the high carcinogenic risk of rice intake for adults and children, respectively. Based on the estimated probability distribution, the probabilities of the total cancer risk (TCR) of rice intake for children from RCS were lower than that from CRC.

## 1. Introduction

Food security is an essential guarantee for world peace and development, which is second only to national security. Soil toxic elements contamination seems to be the greatest challenge for food safety by both reducing crop yields and causing unsafe grain, which has drawn global attention [1]. As in China, a national survey covering more than 70% of the land area has reported that 19% of agricultural soil was contaminated with organic and inorganic contaminants [2]. Most contaminants can be transferred to plants and pose potential risks to human health through the food chain. A high content of toxic elements in soil-borne foods, such as rice and wheat grains, has been reported frequently in recent years [3,4].

Integrated aquaculture-agriculture technologies have shown great potential for food security and poverty alleviation due to their high synergistic effects [5]. Rice and crayfish are the two most popular food species in Asia countries, especially in China. According to the data from the National Bureau of Statistics of China, the production of rice and crayfish was 209.61 million tons and 2.09 million tons, respectively, with a crop area of 29.69 million hectares and 1.29 million hectares in 2019, respectively [6]. In order to improve the yield and profit of crayfish, the rice-crayfish system has been promoted in China over the last three decades. Chinese rice-crayfish system has changed from traditional small-scale farming to large-scale modern farming. In 2019, the area of rice-crayfish integrated system accounted for approximately 85% of the total crayfish farming area [1]. The average productivity of the rice-crayfish integrated system was 45,000 RMB/hm^2^, which was significantly higher than that of the “rice-rape rotation” or “rice-wheat rotation” model [7].

The rice-crayfish system (RCS) is characterized by improving socio-economics and environmental sustainability, i.e., reducing the emission of N_2_O and NH_3_ [8], decreasing the use of chemical fertilizers [9], improving soil quality [10], and enhancing nitrogen use efficiency [11], and has been regarded as a green and sustainable production system in many countries. However, due to the high-profit margin of crayfish, as well as the lack of standard technology, more and more farmers tend to feed excessively, which leads to serious environmental risks, such as cross-contamination of pesticide residues and antibiotics between rice and crayfish, that may occur in this system [12,13].

Several studies have focused on contamination by toxic elements in soil-rice systems [6]. Previous studies have indicated soil pH was the most important factor for toxic elements’ transfer and accumulation in soil-rice systems. Tan et al. estimated the human health risk of crayfish consumption by analyzing a daily intake risk model [6]. Levels of toxic elements in wild crayfish and cultured crayfish were also reported [14]. To our knowledge, information on contamination by toxic elements in paddy soil from the rice-crayfish system is still limited, especially in China, where the area of the rice-crayfish system has rapidly been developed in recent years.

Anhui Province, located in the middle reach of the Yangtze River, is the second-largest area for crayfish production in China, with the rice-crayfish co-culture system area of 0.28 million hectares in 2019. Meanwhile, water pollution in this region, caused by agricultural activity, has received considerable attention [15]. Therefore, the goals of this paper were to quantify the content of toxic elements in soil, rice, and crayfish, as well as undertake a health assessment for the presence of toxic elements. The specific objectives of this study were, therefore, to (1) quantify the content of toxic elements in soil from the rice-crayfish system; (2) identify the sources of toxic elements from soil columns from different culture systems; (3) assess the health impact of toxic elements from agricultural food consumption.

## 2. Materials and Methods

### 2.1. Study Area and Experiment Design

This field experiment was conducted on an experimental farm (29°55′ N, 116°26′ E) located in Anqing City, Anhui Province, covering 9000 ha (Figure 1). The rice-crayfish co-culture system has been practiced on this farm since 2015, and the area of the rice-crayfish co-culture system was 1200 ha in 2019. This region is located in the subtropical monsoon humid climate zone. The annual average rainfall is between 1300 and 1500 mm, the average temperature is 14.5–16.6 °C, and the frost-free period is 248 days.

All field experiments were conducted based on technical specifications for integrated rice-fishing planting and cultivation from China’s Ministry of Agriculture (SC/T 1135.4-2020). Briefly, freshwater crayfish (*Procambarus clarkii*) and rice (*Oryza sativa*) were selected for co-culture in this study. The experiment plot was divided into a small area, with 20 m (L) × 12 m (W), and rice cultivation occupied about 60% of the total area of the plot; the depth of the ditch was 1.0 m. Rice was transplanted in May and harvested in October [16].

### 2.2. Sample Collection and Analysis

A total of 162 soil samples (158 top-layer soil samples and 2 soil-column samples) were collected from the experimental farm in 2019. The soil samples were performed according to ISO 10381-1 (2002). Briefly, the five subsamples were collected from the top layer with a depth of 0-20 cm and then homogenized to one topsoil sample for each sampling site. After collection, all of the soil samples were air-dried at ambient temperature and ground, passing a 2 mm sieve after removing large particles and plant residues. Eventually, all the samples were stored in a sealed plastic bag before analysis. The microwave-assisted, acid digestion was performed. Each soil sample was weighted and digested by HNO_3_-HClO_4_-HF methods in digestion vessels for toxic elements analysis [17]. The content of toxic elements was quantified by an inductively coupled plasma optical emission spectrometer (ICP-OES, Optima 7300 DV, PerkinElmer, Waltham, MA, USA). Soil pH was measured in 1:5 (*w*/*v*) 0.01 M CaCl_2_ soil suspension (ISO 10390:1994). The results were an average of three replicates.

The rice grain samples were ground to pass through 100 mesh after being oven-dried at 105 °C for 10 h and then stored in closed polyethylene bags. All rice samples were determined by an inductively coupled plasma optical emission spectrometer (ICP-OES) after digestion using HNO_3_ and H_2_O_2_. The microwave digestion procedure and instrumental conditions of ICP-OES are listed in Appendix A.

The analytical calibration curves were developed from eight standard solutions ranging from 0.01 mg L^−1^ to 10 mg L^−1^. The limits of detection (LOD) and quantification (LOQ) of the method were obtained by standard deviations of 10 blanks and background equivalent concentration (BEC). For the quality assurance and quality control, blank, duplicate samples, and reference materials (GBW07403 for soil samples, GBW10010 for rice samples, GBW10024 for crayfish samples) were used to verify the accuracy and precision of the digestion and analysis process. The accepted recovery rate ranged from 80% to 120%. The relative deviation of the duplicate samples was less than 7% for all treatments. The limits of detection (LOD) for As, Cd, Ni, Cu, Pb, Hg, Zn and Cr were 0.01, 0.01, 1.0, 1.0, 1.0, 0.002, 1 and 2 mg·kg^−1^, respectively. All analyzed results were calculated on a dry-weight basis.

### 2.3. Health Risk Assessment Methods

To sufficiently describe the overall human health risk of toxic elements, human non-carcinogenic and carcinogenic risk assessments were used to evaluate the risk through consumption of rice and crayfish by the adults and children, according to the methods by USEPA [18].

#### 2.3.1. The Estimated Daily Intake (EDI)

In this study, the EDI (μg kg^−1^ day^−1^) was used to determine the dietary exposure of adults and children to elements using the following equation:$$\mathrm{EDI}=\frac{\mathrm{EF}\times \mathrm{ED}\times \mathrm{VI}\times \mathrm{MC}}{\mathrm{BW}\times \mathrm{AT}}\;$$
where EF and ED are the exposure frequency and exposure duration, with the value of 365 days year^−1^, 77 years for adults, and 9 years for children, respectively. VI represent ingestion rate (rice: children 0.24 kg·person^−1^ d^−1^, adults 0.337 kg·person^−1^ d^−1^; crayfish: children 0.02 kg·person^−1^ d^−1^, adults 0.0555 kg·person^−1^ d^−1^) [14]. MC is the content of toxic elements (mg·kg^−1^, dry weight). The BW value is the average body weight (70 kg for adults, 25 kg for children), and AT is equal to ED × 365 days·year^−1^.

#### 2.3.2. Non-Carcinogenic Risk Assessment

The model for estimating HI was determined by the following equation:$$\mathrm{HI}=\sum \mathrm{THQ}=\sum \frac{\mathrm{EDI}}{\mathrm{RfD}}=\sum \frac{\mathrm{EF}\times \mathrm{ED}\times \mathrm{VI}\times \mathrm{MC}}{\mathrm{RfD}\times \mathrm{BW}\times \mathrm{AT}}$$
where hazard index (HI) is the arithmetic sum of the individual metal’s target hazard quotient (THQ); RfD is the reference oral dose of toxic elements (As, 0.3; Hg, 0.16; Cr, 1500; Cu, 40; Ni, 20; Zn, 300; Cd, 1; Pb, 3.5 μg·kg^−1^ day^−1^). Humans may tolerate the potentially toxic effects if the THQ or HI value exceeds 1.0.

#### 2.3.3. Carcinogenic Risk Assessment

The lifetime cancer risk (CR) and total cancer risk (CRt) were used to assess the incremental probability of an individual developing cancer using the following equation:$${\mathrm{CR}}_{T}=\sum \mathrm{CR}=\sum \mathrm{EDI}\times \mathrm{SF}=\sum \frac{\mathrm{EF}\times \mathrm{ED}\times \mathrm{VI}\times \mathrm{MC}\times \mathrm{SF}}{\mathrm{BW}\times \mathrm{AT}}$$
where CRT is the sum of cancer risk of specific carcinogenic toxic elements. The cancer slope factor (SF, kg·day mg^−1^) is 15 for Cd, 1.5 for As, 0.5 for Cr, 0.0085 for Pb, and 0.84 for Ni, based on USEPA integrated risk information system and California OEHHA toxicity criteria database. The criteria for risk are as follows: no significant health risk (CR or CRt < 10^−6^); acceptable/tolerable (10^−6^ < CR or CRt < 10^−4^); unacceptable (CR or CRt > 10^−4^) according to USEPA. 

Furthermore, considering the random variations in levels for toxic elements, the traditional deterministic risk assessment method could overestimate or underestimate the risk due to the use of deterministic parameters [19]. To reduce the uncertainty of the health risk of toxic elements, a Monte Carlo simulation was conducted to assess the probabilistic risk using the Crystal Ball program (version 11.1). In this study, the THQ and CR values were repeatedly and randomly calculated for 10,000 iterations. The parameters distribution types used in this study were listed in Appendix A.

### 2.4. Statistical Analysis

Statistical processing of data from the surveyed monitoring plots was performed using statistical functions of STATISTICA 10.0 software (Stat Soft Inc., Tulsa, OK, USA). We used variation statistics methods to calculate the arithmetic mean, minimum and maximum values, and confidence interval of the mean. The approximation of the empirical distribution by distribution laws was performed in the distribution fitting module of STATISTICA 10.0 (Stat Soft Inc., Tulsa, OK, USA). Results were considered statistically significant at *p* ≤ 0.05.

## 3. Results and Discussion

### 3.1. Physical and Chemical Properties of Soil

Soil pH is one of the most important parameters that affect the migration and transformation of toxic elements in soil [20]. In this study, soil pH values varied from 6.61 to 8.31, with a mean value of 7.96, indicating a slightly alkaline nature of the RCS soil. As for the conventional rice culture (CRC), the soil of CRC had higher average pH than that of RCS (*p* < 0.05). The high pH decreased the solubility and speciation of toxic elements, except for anionic species in soil [20,21]. In contrast, some previous studies pointed out that rice-crayfish systems had a significantly higher soil pH (7.32 ± 0.60, mean ± SD) than traditional rice farming methods [10]. These differences were interpreted to be the addition of feed and long-term measures in terms of flood management that affected the soil pH in the rice-crayfish system [22].

The soil size analysis indicated that the percentages of clay, silt, and sand components were 60.07%, 38.32%, and 1.62%, respectively. Additionally, the soil particles primarily belonged to clay-loam soil according to the National standard of classification and codes for Chinese soil (GB/T 17296-2009). Additionally, the high proportion of clay can reduce the bioavailability of toxic elements in soil, due to its small particle providing a high specific surface area [23].

### 3.2. Toxic Element Levels in Rice-Crayfish System

#### 3.2.1. Levels of Toxic Elements in Topsoil

The contents of eight toxic elements in paddy soil were presented in Table 1 and Figure 2. The mean contents of As, Hg, Cr, Cu, Ni, Zn, Cd, and Pb from RCS were 14.98, 0.08, 97.84, 49.93, 48.13, 114.57, 0.38, and 35.97 mg·kg^−1^, respectively. The coefficient of variance (CV) of toxic elements varied from 5.54% to 18.35%. The CV values lower than 10% indicated low variability and reflected the natural source of contamination [24]. In this study, Hg, As, and Cd had high CV values of 18.35%, 13.46%, and 10.77%, respectively, indicating moderate variability in soil. Furthermore, the high CV values of Hg, As, and Cd were related to being highly influenced by human/external activities.

When using the soil background values of Anhui Province and the environmental quality standard for soils in China as the basis for threshold values in soil, the mean values of eight toxic elements were higher than the background values, and they were lower than the grade II national standards of soil heavy metal content in China. Compared with the data report from China and other locations, the content of elements were lower than those reported nationally and globally [25]. The enrichment values of Hg and Cd were the highest among metals in soil, which was consistent with the data from national paddy fields [8]. The high level of Cd might be due to the application of Cd-containing phosphate fertilizer [26] and the emissions of coal-burning Cd [27]. The relatively high content of As was consistent with the report on the toxic elements pollution in farmland across China, which indicated that the concentration trend of As increased rapidly from 2000 to 2019 [27]. The main source of arsenic in soils may be derived from anthropogenic activities including industrial emission, atmospheric deposition, pesticides, and fertilizers [28]. Mercury in soils was related to the atmospheric deposition of fossil fuel combustion [29]. A high level of arsenic can be accumulated in rice grains, especially under anaerobic conditions [30]. Thus, it may cause stress in rice seedlings [31]. The high enrichment of Cu may be due to the many prominent Cu mines in Anqing city [32], and the presence of Cu will alter the genes related to fatty acid metabolism [33]. Additionally, although Ni was an essential micronutrient for normal growth in plants, the high content of Ni found in this study exceeded the permissible limit in soil (35 mg·kg^−1^) [34]. The sequence of the average content of toxic elements in soil from CRC were ranked as Zn > Cr > Cu > Ni > Pb > As > Cd > Hg. In general, the levels of the toxic elements were lower than those of RCS, except Hg.

#### 3.2.2. Vertical Distribution of Toxic Elements in Soil

The vertical distribution pattern of toxic elements in the soil was shown in Figure 3. In general, the order of the mean content values was Zn > Cr > Cu > Ni > Pb > As > Cd > Hg. The vertical distribution of metals in soil is mainly affected by bioturbation, diffusion, etc. [35]. The contents of eight toxic elements, except for Hg, were highest in the topsoil (0–20 cm) from the rice-crayfish system, which could be due to the effect of superficial enrichment through crayfish culture. With the soil depth increased to 20 cm, the contents of most toxic elements decreased. These results indicated that the RCS would promote the accumulation of toxic elements in topsoil. In contrast, the vertical distribution of toxic elements content in soil from conventional rice culture was different. The levels of most metals decreased when the soil depth increased to 40 cm, which might be attributed to the migration capability and mobility of different toxic elements. The contents of Hg and Cd were relatively stable, and Zn had a relatively high content, in the range of 40–60 cm. Citeau et al. [36] reported that Zn was greatly mobile and easily moved down through the soil profile. The high levels of Cr and Ni in the topsoil also indicated that the net input of these metals in soil has increased in recent years, which was consistent with the report on the inventory of trace elements in farmland across China [37]. Therefore, more consideration should be given to Cr and Ni.

The soil pH is the key factor governing the solubility and content of soluble metals [29]. In this study, the pH values of soil from top to bottom were 8.25, 8.49, and 8.32, respectively. The higher pH in the middle soil of the profile would decelerate the leaching of toxic elements, which was consistent with the distribution pattern of toxic elements (Cr, Cu, Ni, Pb, and As).

### 3.3. Toxic Element Levels in Edible Parts

Levels of toxic elements in crayfish and rice samples were listed in Table 2. The As, Cd, Cr, Cu, Hg, Ni, Pb, and Zn mean values in crayfish (dry weight) were 0.134, 0.001, 0.066, 2.623, 0.086, 0.024, 0.035, and 11.767 mg·kg^−1^, respectively (*N* = 16, tail muscle samples). The accumulation of metals in crayfish tissues might be correlated with their omnivory and necrophagia, and therefore, crayfish have been used as bioindicators of contaminants [38]. High contents of Cu and Zn were observed in crayfish muscle, and these two elements have been reported as essential metal cofactors used to maintain normal reproduction [39]. Furthermore, crayfish consumption can offer essential elements for humans [40]. The levels of Cd, Cr, As, and Pb were below the maximum acceptable levels in the Chinese national food safety standard (GB2762-2017) and the WHO, and the levels of toxic elements in this study were much lower than the global data summarized by Kouba et al. [41]. The low levels of toxic elements may be explained by aquaculture management with the characteristics of short cultivation duration and high growth rate [42]. Similar research also indicated that the average concentration of toxic elements in cultured crayfish was lower than that in wild crayfish [14].

As for the toxic elements in grains, the data showed that Zn had the highest in grains (mean value = 11.23 mg·kg^−1^ for RCS, 10.96 mg·kg^−1^ for CRC). After Zn, the sequence of concentrations in grains was Cu > Ni > Cr > As > Pb > Cd > Hg. These values were comparable with a previous study of China and Anhui Province [43].

To properly elucidate the transfer behavior of toxic elements from soil to plant, the transfer factor of soil to rice (TF_rice grain/soil_) was calculated for RCS and CRC systems and is shown in Table 2. The high TF value indicated more toxic elements were taken up by plants. On average, the TF values of As, Cd, Cr, Cu, Hg, Ni, Pb, and Zn from RCS were 0.003, 0.066, 0.001, 0.062, 0.125, 0.004, 0.001, and 0.098, respectively, following the order of Hg > Zn > Cd > Cu > Ni > As > Pb ≈ Cr, which indicated that Hg had highest bioavailability and mobility. In comparison, the mean values of TF for the CRC system were different, among which Zn had the highest transfer factor, followed by Cu, Cd, Hg, As, Ni, Pb, and Cr. These results indicated that there was no significant variation between toxic elements, except Hg. The transfer factor of Hg for RCS was two times higher than that for CRC, indicating that the RCS system favored the transfer of Hg and may cause potential health risks despite the low content of Hg in soil. Zinc and Cu were essential for rice growth. These results were in line with previous observations on rice-soil systems from other regions [44].

### 3.4. Potential Sources of Toxic Elements in Soil

Multivariate statistical methods such as correlation analysis and principal component analysis (PCA) were used to identify the pollution sources of toxic elements in the soil.

#### 3.4.1. Correlation Analysis

Correlation analysis was used to determine the inter-relationships among elements [45]. As shown in Table 3, all of the metal pairs showed positive relations with each other at a 99% confidence level. The Pearson correlation analysis indicated high correlation coefficients between toxic elements As-Cr (0.741), As-Cu (0.761), As-Ni (0.720), Cr-Cu (0.715), Cr-Ni (0.868), Cr-Zn (0.861), Cu-Ni (0.671), Cu-Zn (0.720), and Ni-Zn (0.836) in soil from CRC at 0.05 significant level, and As-Zn (0.914) showed highly significantly positively correlation at 0.05 significant level. Results indicated that these elements were likely derived from the same or similar contamination source. As for the RCS soil, Zn was found to be significantly positively correlated with Cr (0.923), Cu (0.927), and Ni (0.963) (*p* < 0.05). Ni was also observed to be significantly positively correlated with Cr (0.962) and Cu (0.913) at a 0.01 significant level. In general, most elements in RCS soil showed more positive correlations than those from CRC soil, indicating possible same sources for these toxic elements as those from RCS soil.

#### 3.4.2. Principal Component Analysis

The principal component analysis (PCA) was used to identify toxic elements sources based on the statistical data. The results of the PCA analysis for toxic elements in soil from CRC are listed in Table 4, the Kaiser-Meyer-Olki (KMO) value was found to be 0.751 (>0.7), and the Bartlett test of sphericity of significance yielded 0.00 (*p* < 0.05), both indicating the strong correlation among toxic elements and good validity. Based on eigenvalues (eigenvalue > 1), the results showed that three main principal components (PCs) explained 85.44% of the total variance. Based on the component matrix, the first factor (PC1) accounted for 53.00% of the total variance and loaded heavily on Zn, Cr, As, Ni, and Cu. This result was also supported by Pearson’s correlation analysis, indicating that these toxic elements were positively correlated with each other. Furthermore, it may imply a common source of agricultural activities for these metals. The application of fertilizers, pesticides, and manures can promote the accumulation of toxic elements in soil [2]. The second factor (PC2) accounted for 18.71% of the total variance and showed high factor loading values of Pb and Cd, which was attributed to traffic-related activities. The third factor (PC3) was loaded on Hg, accounting for 13.73% of the total variance. As reported, the source of Hg in soil was related to atmospheric deposition from fossil fuel combustion [29].

As shown in Appendix A, the PCA data for RCS indicated that the Kaiser-Meyer-Olki (KMO) value and Bartlett test of sphericity of significance were found to be 0.901 (>0.7) and 0.00 (*p* < 0.05), respectively. Only one principal component (PC1) was extracted, accounting for 71.39% of the total variance. PC1 showed high loading values for Zn, Ni, Cr, Cu, Pb, and As. Moreover, the significant correlations among these metals were also detected by Pearson correlation coefficients. Compared with the data of CRC, toxic elements in RCS soil mainly result from agricultural practices. During the co-culture process, phosphate fertilizer is added to maintain the weight and growth rate of crayfish, which leads to the accumulation of toxic elements. In addition, crayfish aquaculture significantly affects the soil microbial ecological environment by causing a decrease in soil microbial diversity [6,46], which also affects the behaviors of soil toxic elements.

### 3.5. Potential Health Risk Assessment of Toxic Elements

#### 3.5.1. The Estimated Daily Intake

The EDI values of Cr, Ni, Cu, Zn, As, Cd, Hg, and Pb due to the rice consumption for adults from CRC were found to be 0.74, 0.93, 30.69, 105.24, 0.64, 0.22, 0.05, and 0.36 μg kg^−1^ d^−1^, respectively. Additionally, the corresponding EDI values from RCS were 0.33, 0.88, 15.02, 54.06, 0.19, 0.12, 0.05, and 0.17 μg kg^−1^ d^−1^, respectively. The EDI values demonstrated a descending order of Zn > Cu > Ni > Cr > As > Pb > Cd > Hg in all groups. All EDI values were less than the maximum tolerable daily intake (MTDI) prescribed by the WHO [47] for each element, as indicated in Table 5. In this study, the dietary intake of toxic elements for children was higher than that for adults. 

#### 3.5.2. Non-Carcinogenic Risk

Non-carcinogenic risks of toxic elements in rice and crayfish were calculated and are listed in Table 6. As for rice, the THQ values ranked as As > Cu > Zn > Hg > Cd > Pb > Ni > Cr for both adults and children. Additionally, the mean THQ values of As in rice grains from CRC for adults were found to be greater than 1, which indicated obvious effects. It suggested that As was the main element that posed a potential non-carcinogenic risk for local residents. Similar results were also found in the Yangtze River delta area [48] and Fujian Province [49]. The health risk posed by Zn, Hg, Cd, Pb, Ni, and Cr through rice consumption was negligible. For all of the toxic elements, the THQ values for children were higher than those for adults. Compared with adults, children were more susceptible to toxic elements. The hazard indices (HIs) of rice from both CRC and RCS for children and adults exceeded 1.0. Arsenic was the major element contributing to the non-carcinogenic health risk, followed by Cu and Zn. Additionally, the results of Monte Carlo simulations are shown in Figure 4, indicating that As with 64.5% and 99.9% probability had a higher risk than the effect on THQ in rice from CRC for adults and children, respectively. Moreover, the HI values from RSC were lower than those from CRC, which indicated that the RSC could alleviate the non-carcinogenic risk of toxic elements in rice. A similar result was also shown by the Monte Carlo simulations—namely, the probabilities of the potential risk of toxic elements from CRC were higher than those from RCS. Nevertheless, the high HI values demonstrated rice consumption can result in adverse health effects and pose a non-carcinogenic risk in this area.

In terms of crayfish, the values for THQ and HI were lower than the acceptable safe limit (HI < 1), confirming that the normal consumption of toxic elements in crayfish will pose an insignificant risk of non-cancer effects to consumers.

#### 3.5.3. Carcinogenic Risk

In this study, Cd, As, Cr, Pb, and Ni were considered carcinogenic toxic elements. Based on the 10^−4^ threshold, the cancer risk levels of these toxic elements except Pb were above the recommended cancer risk limit, indicating the high carcinogenic health risk from rice consumption, and Cd represented the largest carcinogenic risk factor. As for the ingestion of crayfish, the CR value for As was higher than the 10^−4^ threshold, which meant potential cancer risk to consumers. The cancer risk of As in crayfish has been documented by other authors [42,50]. Only the CR value of Pb was less than 10^−6^, meaning that Pb did not confer carcinogenic risk to adults and children. Additionally, these results were consistent with previous reports [51,52]. In fact, the potential health risk from toxic elements exposure is likely to be greater than the data calculated in this study, as local residents may also be exposed to other types of food contaminated by toxic elements.

The probability distribution of total cancer risk for crayfish is shown in Figure 5, from which the median value was 1.11 × 10^−3^ and 4.66 × 10^−4^ for children and adults, respectively. This also indicated that the total cancer risk from crayfish was higher for children than that for adults.

Figure 6 and Figure 7 illustrated the probabilities of the TCR of rice for adults and children. Generally, the median percentile values of TCR were significantly higher than those of the threshold, suggesting that rice consumption might pose a carcinogenic risk for the residents. As for the different culture systems, the median values of TCR_rice_ for children greatly varied, indicating that the RCS can decrease the carcinogenic risk for children through rice consumption.

## 4. Conclusions

The results obtained in this work showed that the eight toxic elements in soil from CRC and RCS were variously distributed in topsoil and soil column, and the main source of toxic elements from RCS soil was agronomic practices, based on results of correlation analysis and principal component analysis. The levels of toxic elements in rice and crayfish were below the maximum permissible limits, while Hg in rice from RCS showed the highest bioavailability and mobility. Health risk assessment, coupled with Monte Carlo simulation, revealed that dietary exposure to As and Hg from rice and crayfish consumption was the primary factor for non-carcinogenic risk. Cd and As were the dominant contributor to the high carcinogenic risk of rice intake for adults and children, respectively. Furthermore, based on the estimated probability distribution, the probabilities of the total cancer risk of rice intake for children from RCS were lower than that from CRC, indicating that RCS can decrease the possibility of cancer risk due to rice consumption in children.

## Figures and Tables

**Figure 1 foods-11-01160-f001:**
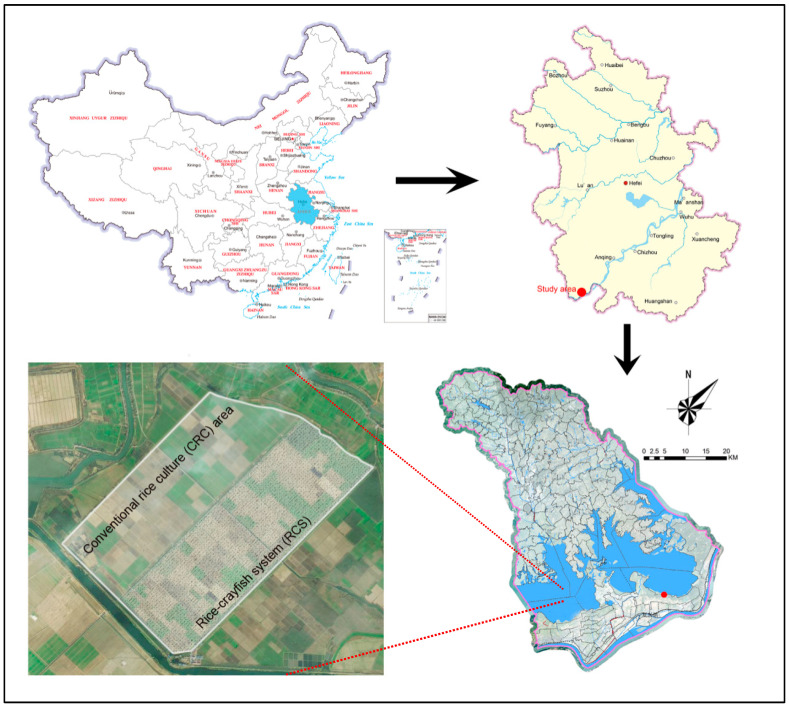
Geographic location of the experimental farm.

**Figure 2 foods-11-01160-f002:**
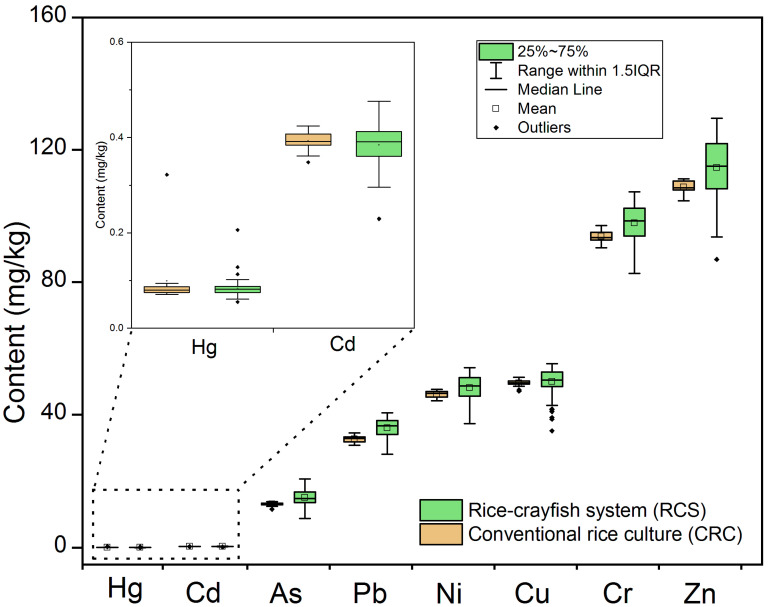
Box plots of content of toxic elements in soil.

**Figure 3 foods-11-01160-f003:**
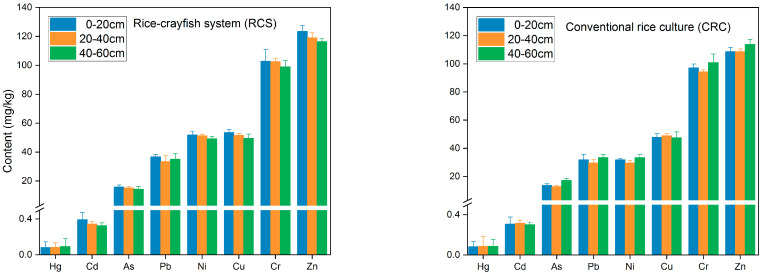
Vertical distribution of toxic elements in soil column.

**Figure 4 foods-11-01160-f004:**
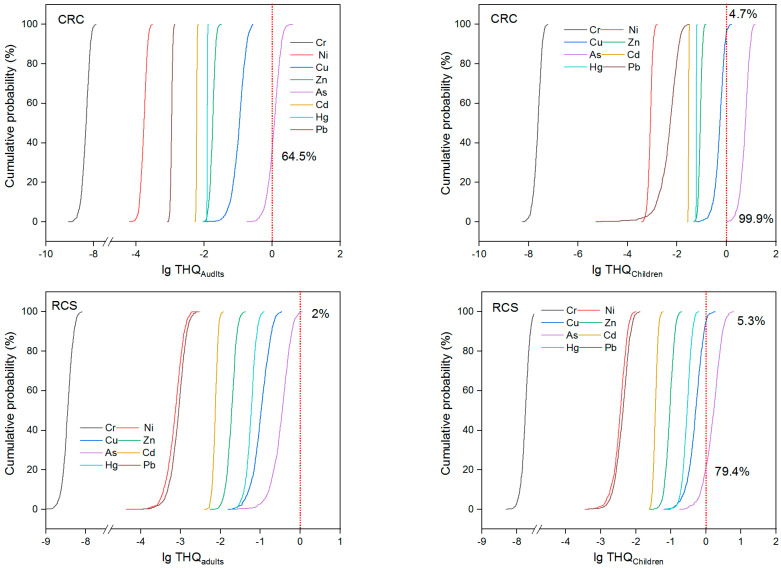
Monte Carlo simulations of THQ cumulative probability for children and adults exposed to toxic elements in rice from CRC and RCS (the red short dot line presents the risk boundary of THQ = 1).

**Figure 5 foods-11-01160-f005:**
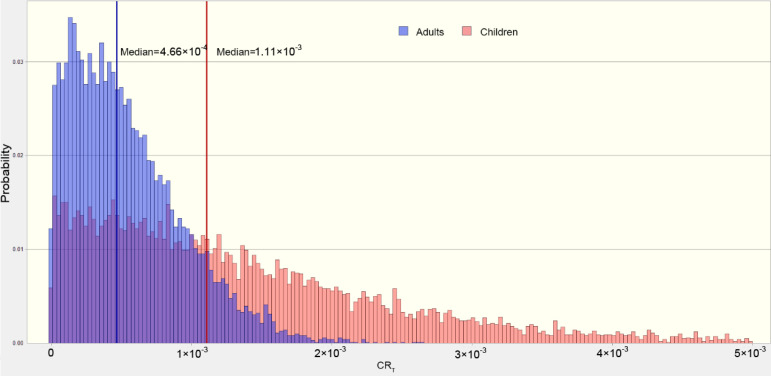
The estimated probability distribution of total cancer risk (CRt crayfish) values for adults and children.

**Figure 6 foods-11-01160-f006:**
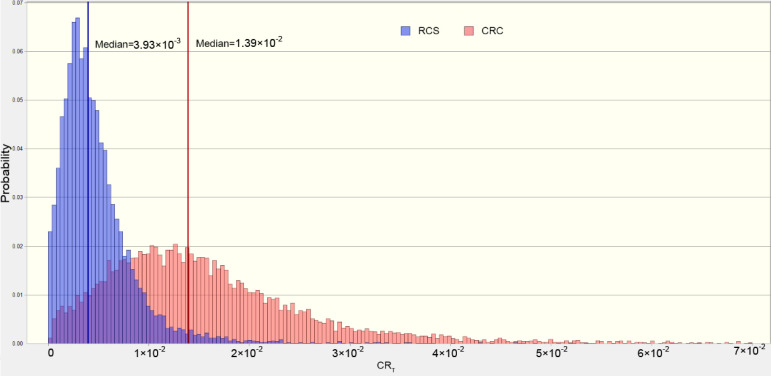
The estimated probability distribution of (CRt rice) values for children.

**Figure 7 foods-11-01160-f007:**
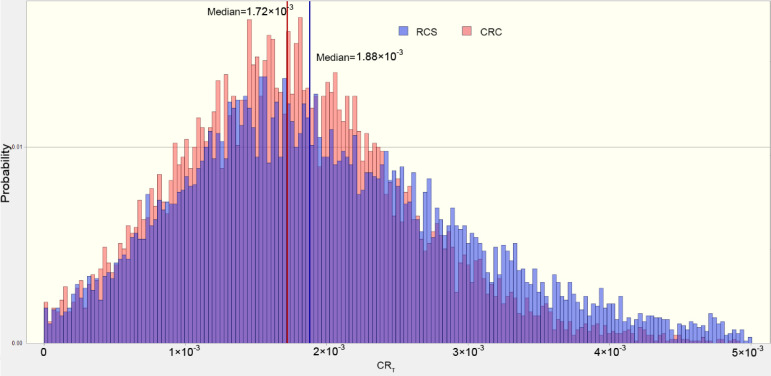
The estimated probability distribution of (CRt rice) values for adults.

**Table 1 foods-11-01160-t001:** Summary statistics for toxic element contents in soil (unit: mg kg^−1^).

		As	Hg	Cr	Cu	Ni	Zn	Cd	Pb
Rice-crayfish system (RCS)	Minimum	8.72	0.06	82.70	35.10	37.30	86.90	0.23	28.00
	Maximum	20.60	0.21	107.30	55.40	54.20	129.50	0.48	40.50
	Median	14.71	0.08	98.50	50.45	48.65	115.10	0.39	36.65
	Mean	14.98	0.08	97.84	49.93	48.13	114.57	0.38	35.97
	SD	2.02	0.02	5.42	3.77	3.62	8.45	0.04	2.59
	CV (%)	13.46	18.35	5.54	7.54	7.51	7.37	10.77	7.19
Conventional rice culture (CRC)	Minimum	11.47	0.07	90.40	47.10	44.20	104.60	0.35	30.70
	Maximum	13.72	0.32	97.10	51.30	47.60	111.20	0.42	34.50
	Median	13.14	0.08	93.50	49.60	46.40	108.50	0.39	32.80
	Mean	12.99	0.10	93.90	49.50	46.11	108.67	0.39	32.60
	SD	0.58	0.07	2.03	1.23	1.14	2.13	0.02	1.07
	CV (%)	0.04	0.68	0.02	0.02	0.02	0.02	0.05	0.03
Background value ^a^		9.00	0.03	66.5	20.4	29.8	62	0.10	26.6
Safety limits ^b^		25	0.5	200	100	90	250	0.45	80
Soils in China ^c^		8.89	0.07	67.37	25.81	27.77	85.86	0.19	30.74

**Notes:** Max and Min are minimum and maximum, respectively; CV represents coefficient of variance; SD represents standard deviation. ^a^ stands for the background value of heavy metals in Anhui Province (Zeng et al. 2011); ^b^ stands for the safety limits for soil by the Chinese Environment Protection Administration (Soil environmental quality—Risk control standard for soil contamination of agriculture land. GB15618-2018); ^c^ stands for the content of heavy metals in soils from China (Yuan et al. 2021).

**Table 2 foods-11-01160-t002:** The levels of toxic elements in crayfish and rice (*n* = 3, on dry weight, mg·kg^−1^).

		As	Cd	Cr	Cu	Hg	Ni	Pb	Zn
Crayfish									
Tail muscle		0.134	0.001	0.066	2.623	0.086	0.024	0.035	11.767
Maximum permissible limits for metals by China EPA		0.15	0.20	1.00	-	0.02	-	0.20	-
Maximum permissible limits for Metals by the WHO		1.00	0.40	-	-	0.02	-	0.20	-
Rice grain									
	RCS	0.040	0.025	0.068	3.120	0.010	0.183	0.035	11.230
	CRC	0.067	0.023	0.077	3.197	0.005	0.097	0.038	10.963
TF_grain/soil_	RCS	0.003	0.066	0.001	0.062	0.125	0.004	0.001	0.098
	CRC	0.005	0.059	0.001	0.065	0.050	0.002	0.001	0.101

**Note:** TF_grain/soil_ was defined as the ratio of the toxic element content in the rice grain to the toxic element content in soil. ‘-’: No corresponding value was set.

**Table 3 foods-11-01160-t003:** Correlation coefficients between toxic elements and the soil pH from RCS.

**Soil from CRC (Conventional Rice Culture)**									
	**pH**	**As**	**Hg**	**Cr**	**Cu**	**Ni**	**Zn**	**Cd**	**Pb**
pH	1								
As	−0.101	1							
Hg	−0.069	0.231	1						
Cr	−0.072	**0.741 ****	−0.148	1					
Cu	−0.207	**0.761 ****	−0.086	**0.715 ****	1				
Ni	−0.233	**0.720 ****	−0.021	**0.868 ****	**0.671 ***	1			
Zn	−0.063	**0.914 ****	0.079	**0.861 ****	**0.720 ****	**0.836 ****	1		
Cd	−0.336	0.141	−0.017	0.173	0.293	0.094	0.267	1	
Pb	0.148	0.223	0.244	0.122	0.307	−0.121	0.254	0.427	1
**Soil from RCS (Rice–crayfish system)**									
pH	1								
As	0.146	1							
Hg	−0.094	0.273 **	1						
Cr	−0.139	**0.788 ****	0.335 **	1					
Cu	−0.130	**0.748 ****	0.375 **	**0.884 ****	1				
Ni	−0.099	**0.826 ****	0.386 **	**0.962 ****	**0.913 ****	1			
Zn	−0.090	**0.803 ****	0.411 **	**0.923 ****	**0.927 ****	**0.963 ****	1		
Cd	0.086	0.318 **	0.292 **	0.335 **	0.532 **	0.406 **	0.510 **	1	
Pb	−0.134	**0.785 ****	0.427 **	**0.848 ****	**0.793 ****	**0.864 ****	**0.856 ****	0.325 **	1

* Correlation is significant at the 0.05 level (2-tailed). ** Correlation is significant at the 0.01 level (2-tailed). Bold: significant correlation.

**Table 4 foods-11-01160-t004:** The total variance explained and component matrices for the toxic elements in soils from CRC.

**Component**	**Initial Eigenvalues**			**Extraction Sums of Squared Loadings**			**Rotation Sums of Squared Loadings**			
	**Total**	**% of Variance**	**Cumulative %**	**Total**	**% of Variance**	**Cumulative %**	**Total**	**% of Variance**	**Cumulative %**	
1	4.240	53.005	53.005	4.240	53.005	53.005	4.092	51.156	51.156	
2	1.497	18.709	71.714	1.497	18.709	71.714	1.559	19.484	70.640	
3	1.098	13.728	85.442	1.098	13.728	85.442	1.184	14.802	85.442	
4	0.560	6.998	92.441							
5	0.307	3.836	96.276							
6	0.204	2.554	98.831							
7	0.081	1.008	99.839							
8	0.013	0.161	100.000							
**Metals**	**Component Matrix ^a^**						**Rotated Component Matrix ^a^**			
	**PC1**		**PC2**	**PC3**			**PC1**	**PC2**		**PC3**
Zn	0.961		0.002	0.079			0.940	0.186		0.101
Cr	0.911		−0.215	−0.102			0.931	−0.143		−0.072
As	0.909		0.030	0.268			0.925	0.069		−0.160
Ni	0.871		−0.358	0.073			0.899	0.113		0.281
Cu	0.860		0.058	−0.159			0.812	0.319		−0.088
Pb	0.264		0.846	−0.090			0.124	0.832		−0.175
Cd	0.305		0.612	−0.521			0.067	0.831		0.314
Hg	0.036		0.478	0.837			−0.004	0.044		0.963

**Note:** Extraction method: principal component analysis; rotation method: varimax with Kaiser normalization; ^a^ rotation converged in 4 iterations.

**Table 5 foods-11-01160-t005:** The estimated dietary intake (EDI) of toxic elements (μg kg^−1^ d^−1^) via consumption of rice and crayfish.

			Cr	Ni	Cu	Zn	As	Cd	Hg	Pb
RCS	Rice	Children	0.65	1.76	29.95	107.81	0.38	0.24	0.10	0.34
		Adults	0.33	0.88	15.02	54.06	0.19	0.12	0.05	0.17
	Crayfish	Children	0.05	0.02	2.10	9.41	0.11	0.00	0.07	0.03
		Adults	0.05	0.02	2.08	9.33	0.11	0.00	0.07	0.03
CRC	Rice	Children	0.74	0.93	30.69	105.24	0.64	0.22	0.05	0.36
		Adults	0.37	0.47	15.39	52.78	0.32	0.11	0.02	0.18
Maximum tolerable daily intakes (MTDI) [47]			300	12	500	300	2.14	0.8	0.23	1.5

**Table 6 foods-11-01160-t006:** Non-carcinogenic risk for toxic elements in rice and crayfish (mean value).

			THQ								HI
			Cr	Ni	Cu	Zn	As	Cd	Hg	Pb	
Rice	CRC	Adults	0.0002	0.023	0.38	0.17	1.07	0.11	0.15	0.052	1.97
		Children	0.0005	0.046	0.77	0.35	2.13	0.22	0.29	0.10	3.91
	RCS	Adults	0.0002	0.044	0.37	0.18	0.63	0.12	0.29	0.048	1.70
		Children	0.0004	0.088	0.75	0.36	1.26	0.24	0.59	0.095	3.38
Crayfish	RCS	Adults	<0.0001	0.0009	0.052	0.031	0.35	0.001	0.43	0.0079	0.87
		Children	<0.0001	0.0009	0.053	0.031	0.36	0.001	0.43	0.0080	0.88

## Data Availability

The data that support the findings of this study are available from the corresponding author upon reasonable request.

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
