# Peer review of "A Multi-Medium Analysis of Human Health Risk of Toxic Elements in Rice-Crayfish System: A Case Study from Middle Reach of Yangtze River, China"

_foods, 2022, doi:10.3390/foods11081160_

Round 1
Reviewer 1 Report
- Give the reference for heavy metal analysis in Methodology. line 99
- Write the full form of ICP-OES. Line 106
- Table 1 is not listed in the manuscript, but only explained in the text line 148.
- What are the effect that can cause arsenic metal? Give reasons
- Give the expansion for all the metals in the text.
Reviewer 2 Report
Dear, considering submitted manuscript for rewiev, my opinion are as follows. The manuscript presented by the authors is interesting, well organized, and well written. However, the manuscript must be improved before publication. The main suggestions are inserted in comments. The main shorthcomin is related to methodology used for the health risk assessment, followed by results presented in the tables and figures. Therefore, methodology must be improved and then results presented in tables and figures must be improved accordingly.

Reviewer 3 Report
The Manuscript entitled “A multi-medium analysis of human health risk of heavy metals in rice-crayfish system, a case study from middle reach of Yang-tze River, China”, is an interesting work. However, in my opinion, some aspects should be improved before being considered for their publication.
Page 1. Line 41, Introduction. Please revise and modify the use of tons and tones.
Page 2, line 56, Introduction. Please verify the spaces between the references in all the manuscript, for example “[12,13]”.
Page 3, lines 91-102, Sample collection and analysis. Please add the amount of sample used in the process.
Page 3, lines 91-102, Sample collection and analysis. Please describe in detail the digestion process, (amount of acid, sequence, time, batch, or microwave), and add a reference that sustains the process.
Page 4, lines 107-113. Sample collection and analysis. Please provide the analysis condition by ICP-OES.
Page 4, lines 107-113. Sample collection and analysis. Please provide the analytical parameters by ICP-OES, (linear regression, the LOD, LOQ…).
Page 4, lines 107-113. Sample collection and analysis. Please check and modify the paragraph “For the quality assurance… the digestion and analysis process.” is confusing. Additionally explain the term blank reagent, is referred to a blank sample?
Page 4. Line 137. Carcinogenic risk assessment. Please check and homogenize the concentration units in the manuscript (kg day/ mg or kg day mg-1).
Page 5, lines 158-171. Results and discussion. Please add a table or graphic where the parameters such as migration and transformation of a heavy metal show their possible affectations in the function of the pH value (6.61 to 8.31).
Page 5, lines 158-171. Results and discussion. Based on the following paragraph “In contrast, some previous studies pointed…. traditional rice farming methods [10] Please add the pH value reported in the literature.
Page 5, lines 158-171. Results and discussion. Please add the number of samples employed to determine the results in table 1 and the corresponding %RSD values. I suggest modifying the position in section 3.2.1.
Page 6, lines 195-196. Content of heavy metals in top soil. According to the following paragraph “The enrichment of Hg and Cd was the highest among metals in soil, which was consistent with the data from national paddy fields”. Please explain in detail the cause or causes to obtain high levels of Hg in the samples.
Page 8, table 2. Please provide the number of samples analyzed (n=?), %RSD, and the amount of sample employed during the analysis.
Page 9, lines 293-295. Principal component analysis. Please check the use of significant numbers (85.44%, 53%).
Table 5. Please check the use of significant numbers.
Round 2
Reviewer 2 Report
Dear, considering manuscript submitted for second round of review, my opinion is that authors significantly improved manuscript, therefore in such form is acceptable for publication.
Sincerely
Reviewer 3 Report
The Manuscript entitled “A multi-medium analysis of human health risk of heavy metals in rice-crayfish system, a case study from middle reach of Yang-tze River, China”. In my opinion, the current version of the manuscript can be considered for their publication.